# Nanoparticles in 472 Human Cerebrospinal Fluid: Changes in Extracellular Vesicle Concentration and miR-21 Expression as a Biomarker for Leptomeningeal Metastasis

**DOI:** 10.3390/cancers12102745

**Published:** 2020-09-24

**Authors:** Kyue-Yim Lee, Ji Hye Im, Weiwei Lin, Ho-Shin Gwak, Jong Heon Kim, Byong Chul Yoo, Tae Hoon Kim, Jong Bae Park, Hyeon Jin Park, Ho-Jin Kim, Ji-Woong Kwon, Sang Hoon Shin, Heon Yoo, Changjin Lee

**Affiliations:** 1Department of Cancer Control, National Cancer Center Graduate School of Cancer Science and Policy, Goyang 10408, Korea; 70564@ncc.re.kr (K.-Y.L.); 75262@ncc.re.kr (J.H.I.); 2Department of Cancer Biomedical Science, National Cancer Center Graduate School of Cancer Science and Policy, Goyang 10408, Korea; 97055@ncc.re.kr (W.L.); jhkim@ncc.re.kr (J.H.K.); yoo_akh@ncc.re.kr (B.C.Y.); 71917@ncc.re.kr (T.H.K.); jbp@ncc.re.kr (J.B.P.); 3Center for Pediatric Cancer, National Cancer Center, Goyang 10408, Korea; hjpark@ncc.re.kr; 4Department of Neurology, National Cancer Center, Goyang 10408, Korea; hojinkim@ncc.re.kr; 5Neuro-Oncology Clinic, National Cancer Center, Goyang 10408, Korea; jwkwon@ncc.re.kr (J.-W.K.); nsshin@ncc.re.kr (S.H.S.); heonyoo@ncc.re.kr (H.Y.); 6Rosetta Exosome Inc., Seongnam 13494, Korea; clee@rosettaexosome.com

**Keywords:** biomarker, cerebrospinal fluid, extracellular vesicle, microRNA, leptomeningeal metastasis

## Abstract

**Simple Summary:**

Leptomeningeal metastasis (LM) is a terminal stage cancer manifestation to whole neuraxis via cerebrospinal fluid (CSF). Up to now, LM has no solid biomarkers for disease progression or treatment response. Extracellular vesicles (EVs) in biofluids have been recently studied to evaluate cancer diagnostics and prognostics. Here, we measured nanoparticles in human CSF from 472 patients with both Dynamic Light Scattering and Nanoparticle Tracking Analysis. We found that the size distribution and concentration of nanoparticles in LM-disseminating CSF were significantly different from those in non-LM CSF samples. Changes in EVs concentration showed a potential biomarker for the therapy response in patients undergoing intra-CSF chemotherapy. Our suggestion of combined biomarker of EVs concentration and onco-miR for LM chemotherapy could help physicians to perform this possible neurotoxic treatment with appropriate monitoring tools for the effectiveness.

**Abstract:**

Leptomeningeal metastasis (LM) has a poor prognosis and is difficult to diagnose and predict the response of treatment. In this study, we suggested that the monitoring of changes in the concentration of extracellular vesicles in cerebrospinal fluid could help diagnose or predict outcomes for LM. We measured nanoparticles in 472 human cerebrospinal fluid (CSF) from patients including LM with both Dynamic Light Scattering (DLS) and Nanoparticle Tracking Analysis (NTA) after two-step centrifugations. NTA revealed that the concentration of CSF nanoparticles was significantly increased in LM compared to other groups (2.80 × 10^8^ /mL vs. 1.49 × 10^8^ /mL, *p* < 0.01). Changes in NTA-measured nanoparticles concentration after intra-CSF chemotherapy were further examined in 33 non-small cell lung cancer patients with LM. Overall survival was longer for patients with increased EV than the others (442 vs. 165 days, *p* < 0.001). Markers of extracellular vesicles (CD9/CD63/CD81) significantly decreased in the EV-decreased group. MicroRNA-21 expression decreased in this favorable prognostic group, whereas it increased in the EV-decreased group. In conclusion, the elevated concentration of extracellular vesicles in cerebrospinal fluid in patients with LM may be a predictive marker for survival duration. Moreover, EV changes combined with microRNA-21 might be a biomarker for monitoring the efficacy of intracranial chemotherapy of LM in non-small cell lung cancer patients.

## 1. Introduction

Cerebrospinal fluid (CSF) circulates in the central nervous system (CNS) and transports neurotransmitters, hormones, and other actively or passively secreted metabolites from brain cells [1]. Analyzing CSF could detect CNS disease activities such as brain tumors [2,3,4]. Recent studies have examined nanoparticles in human biofluids and their role in normal physiology and disease processes [5,6]. In particular, extracellular vesicles (EVs) are one of the most abundant nanoparticles in biofluids, appearing to stably maintain internal metabolites [7,8]. Human CSF has been studied to be rich in nanoparticles, including EVs and extracellular RNAs [7,8,9]. However, a behavior of EVs in CNS tumor and metastases is not well known due to low level of these materials and a hardship to acquire CSF samples [10,11,12].

Leptomeningeal metastasis (LM) is a terminal-stage cancer that is devastating to patients, with cancer cells spreading through the CSF and adhering to the entire CNS. The overall survival of LM patients is approximately 6–8 weeks, and there is no definitive treatment for LM except intra-CSF chemotherapy, which has a questionable cost/benefit profile due to both low response rate and neurotoxicity [13,14]. Although an early diagnosis is needed to challenge this formidable disease [15,16], a cytology-based diagnosis of LM has a 40–50% false negative rate due to the paucity of floating cancer cells within a small volume (<5 mL) of CSF sample [17,18]. Unlike parenchymal brain metastases, LM has no solid biomarker to monitor disease progression or treatment response, as cancer cells present not as a mass but in a sheet-like linear nodular pattern on neuroimaging [19], and cytology-negative conversion is either rarely achieved or uncorrelated with prognosis [20,21,22]. Therefore, discovering an LM-specific and robust biomarker is needed.

There have been several studies to establish these nanoparticles as a biomarker for CNS tumors, but this has yet to be established mainly due to lack of general existence, reference values from many samples, and difficulty in setting standard method to measure quantitatively. Here, we measured nanoparticles in human CSF from 472 controls and patients with various CNS diseases to provide a reference value and to testify nanoparticles as a biomarker for LM diagnosis and treatment response monitoring. We also tracked changes in EVs concentration in CSF from LM patients after intraventricular chemotherapy and analyzed its relationship with overall survival. Furthermore, we measured the expression of a well-known onco-miRNA, miR-21, in CSF from non-small cell lung cancer patients with LM after intraventricular chemotherapy.

## 2. Results

### 2.1. Patient Characteristics

Demographic characteristics of the 472 patients in the proportion analysis are summarized in Table 1 according to patient group. The median age of all patients was 48 years (range, 0.2–90 years). Non-small cell lung cancer (NSCLC) was the most frequent primary cancer type among LM and brain metastasis patients.

### 2.2. Nano-Sized Particle Peaks in CSF Observed by Dynamic Light Scattering (DLS)

Relative proportions of nanoparticles in CSF samples by Dynamic Light Scattering (DLS) based on intensity particle size distribution (PSD) were obtained using Zetasizer Nano and built-in software. Although the Zetasizer Nano can detect PSD at the millimeter range, we focused on intensity peaks in the 1–1000 nm range for the relative proportion calculation. The majority of the samples (n = 373, 79%) exhibited two peaks of various PSD proportion (Figure 1A,B). A minority of the samples showed a single or triple intensity peak (8% or 13%). For samples that exhibited two peaks, the estimated sizes of CSF particles in the small and the large peaks were 10.5 nm (standard deviation (SD), 4.52) and 176 nm (SD, 179.6), respectively (Figure 1C,D).

The proportions of nanoparticle peaks in CSF varied across individual patients and among patient groups. For samples that exhibited two peaks, the relative proportions of the small and the large peaks differed depending on patient group (Appendix A). Compared with all other patient groups, LM patients exhibited a significantly lower small peak proportion (35% vs. 55%) and higher large peak proportion (64% vs. 44%, *p* < 0.0001). Upon further analysis, we found that the large peak was associated with a significantly larger particle size in LM patients than in the other patient groups (mean 252 vs. 188 nm, *p* < 0.0001; Appendix A).

### 2.3. Verification of Exosomes in CSF 

Based on previous studies [6,7,10], we assumed that extracellular vesicles (EVs) in CSF account for large nanoparticles observed in our CSF samples. To understand the type and origin of EVs, the expression level of several cellular markers was detected in concentrated CSF (Figure 1E). CSF samples were free with cellular proteins such as GM130 and cytochrome c, indicating that the EVs markers present in CSF were not originated with cells contaminating CSF samples. Instead, CD81 and CD63, markers of exosome, were clearly detected. Then, we isolated intact EVs from CSF samples to examine their morphology and sizes. Transmission electron microscopic analysis of EVs isolated from CSFs revealed that EVs in CSF exhibited typical shapes of membranous nanovesicles secreted from mammalian cells, and their sizes were highly heterogeneous ranging from 50 to 200 nanometers in diameter (Figure 1F).

As an indirect method, we observed that the use of a commercially available exosome purification kit that is based on the size exclusion chromatography technology and eliminates protein or nucleic acid from samples (Exo-spin™) nearly abolished the small peak and increased the proportion of the large peak in DLS measurement (Appendix A).

### 2.4. Differences in EV Concentration and Size in CSF Measured by Nanoparticle Tracking Analysis (NTA)

After removing cellular components and fragments, we measured the concentration and size of nanoparticles (presumed as EVs) in CSF from different patient groups by Nanoparticle Tracking Analysis (NTA) using NanoSight NS300 at the same camera condition as described in the Methods. The measured EVs concentration of 472 samples was a mean of 3.46 × 10^8^ particles/mL (SD, 0.29 × 10^8^) and different among patients groups (Figure 2A,B). Healthy control patients (HC) exhibited the lowest EV concentration, with a mean of 2.22 × 10^8^/mL (SD, 159 × 10^8^). Cancer control (CC) and brain metastasis (BM) patients exhibited mean EV concentrations of 2.68 × 10^8^/mL (SD, 6.41 × 10^8^) and 2.49 × 10^8^/mL (SD, 3.15 × 10^8^), respectively. LM patients exhibited the highest mean EV concentrations of 7.15 × 10^8^/mL (SD, 9.15 × 10^8^), which was significantly higher than all the other groups (*p* < 0.0001) except the other CNS disease group. Patients with other CNS diseases (OD) showed the widest distribution of EV concentrations. The OD group consisted of patients with autoimmune disease (n = 25 out of 27 patients with multiple sclerosis), intracranial hemorrhage (n = 13), or CNS infection (n = 9). Patients with autoimmune disease exhibited a mean EV concentration of 1.31 × 10^8^/mL (SD, 1.33 × 10^8^), whereas patients with intracranial hemorrhage or CNS infection exhibited higher mean EV concentrations of 15.32 × 10^8^/mL (SD, 24.15 × 10^8^) and 14.86 × 10^8^/mL (SD, 27.58 × 10^8^), respectively (Appendix A).

We also analyzed the size of EV measured by NTA according to patients groups. The mean EV size of the LM group was significantly larger than other groups (Figure 2C and Appendix A, *p* < 0.001). The mean EV size of the other CNS disease group was also significantly smaller than other groups (*p* < 0.001). In detail, the subgroup of intracranial hemorrhage and autoimmune disease (171 ± 36 nm and 175 ± 22 nm, respectively) drove the EV size of OD groups into the smallest (Appendix A).

To compare the relative EV size distribution between groups, empirical cumulative distribution function (ECDF) plots were generated. LM groups showed the largest distribution as depicted by ECDF plot (Figure 2D), followed by the OD group. As depicted on the ECDF plot, the heterogeneity of the EV size depends on the increase of large size EV in LM patients, whereas that of the OD group came from the relatively small sized EVs. In addition, we analyzed the proportion of EV size with an interval of 50 nm from 0 to 300 nm, which was a tentative limitation of the EV size range (Figure 2E). The LM group showed a significantly higher proportion of EV size more than 150 nm compared to the brain tumor (BT) and control groups (*p* < 0.05, Kruskal-Wallis test, Dunn’s Multiple Comparison method). Meanwhile, the proportion of 50–150 nm sized EV was higher in BT and control groups than that of LM (*p* < 0.05). 

We compared these values by NTA to those measured by DLS. In DLS measurement, we assumed the large peak to be EV-sized and calculated a mean of the large peak size by intensity PSD to be a mean EV size. The discrepancy of the mean values between DLS and NTA was various among patients groups (Appendix A).

### 2.5. Verification of Non-Vesicular Particles among NTA Measured EV in CSF

Although the protein level is relatively low in CSF compared to serum, it has been known that various lipoproteins could be observed at the EV size range in human biofluids [23]. To identify protein aggregates in CSF nanoparticles, we treated CSF with proteinase K (2 ng/mL), which was incubated for 60 min at 37 °C. All samples showed the shift-to-left pattern of peaks above 250–300 nm, but the EV concentration after proteinase is varied (Appendix A). Fourteen (64%) samples showed decreased EV concentration, whereas six samples (27%) revealed increased EV concentration, and two samples had no discernible change (<20%). HC and BM showed mean EV concentration ratios (proteinase K treated/untreated) of 0.98 (±0.32) and 0.97 (±0.70), respectively. However, the LM and OD groups revealed significantly decreased EV concentration ratios of 0.69 (±0.28) and 0.72 (±12.4), respectively (Appendix A).

### 2.6. Change in EV Concentration in LM Patients after Intraventricular Chemotherapy

Among LM patients, 41 were enrolled in a prospective clinical trial (http://cris.nih.go.kr, Identifier: KCT0000082) of ventriculolumbar perfusion (VLP) chemotherapy with methotrexate [24]. Briefly, 24 mg of methotrexate premixed with artificial CSF was continuously infused to the lateral ventricles, and lumbar drainage was used to drain the CSF at the same infusion rate by hydrostatic pressure for 3 consecutive days. Abiding by the protocol, these patients had matched pre-treatment (day 0) and post-treatment (day 4) CSF samples, and we analyzed changes in EV concentration after intraventricular chemotherapy (Appendix A). We found that EV concentration decreased in 25 patients (61%), did not change (<20%) in five patients (12%), and increased in 11 patients (27%) (Appendix A). We evaluated the relationship between treatment-induced change in EV concentration and overall survival. The median overall survival (OS) of patients with increased EV concentration was 342 days (95% confidence interval (CI), 172–512), and it was significantly prolonged to compared the patients with ‘no change (<20% of EV concentration)’ (median 216 days, 95% CI, 117–315) and the patients with decreased EV concentration (median 119 days, 95% CI, 90–148, *p* < 0.05, Figure 3A). Next, to eliminate different primary cancers of various prognosis, we did a subgroup analysis of 33 patients with the same primary cancer of NSCLC, adenocarcinoma. The median OS was 342 days (95% CI, 165–519) in patients with an increased EV concentration (n = 10), but it was only 170 days (95% CI, 97–242) in patients with ‘no change’ (n = 4) and 104 days (95% CI, 76–132) in patients with a decreased EV concentration (n = 19). The difference of OS was more significant in this subgroup analysis compared with that in all patients (*p* < 0.001, Figure 3B). Taken together, we can assume that the change of CSF EV concentration might be a prognostic biomarker in the LM patients who received the VLP with methotrexate treatment.

### 2.7. Analysis of Exosome Surface Markers among NTA Measured EV in Patients with LM Receiving Intraventricular Chemotherapy

As we tentatively defined NTA measured nanoparticles to EVs in this study without exosome extraction process, we were inevitably included non-exosomal particles among EVs. Thus, we verified exosome concentration change using surface markers in selected (remained CSF is enough for further study) patients among those with increased and decreased EV concentration of the above survival analysis [24].

Beads bearing each CD9/CD63/CD81 antibody-capturing exosome in CSF were measured regarding their mean fluorescence intensity (MFI) in patients with increased (*n* = 10) and decreased (*n =* 19) EV concentrations (Figure 4A,B). The MFI values of each sample were converted into a ratio of post-treatment to pre-treatment paired samples. In the decreased group, the EV markers were significantly reduced after intraventricular chemotherapy (ratio 0.64, *p* < 0.001), whereas the increased groups showed no significant exosome concentration change (ratio 1.13).

We also verified these exosome changes further using ExoView Tetraspanin ChipTM in a limited number of CSF samples from the EV-increased and EV-decreased groups (n = 3 from each, Figure 4C,D). A triplicate of each sample revealed that each exosome marker (anti-CD9/CD63/ CD81) after the intraventricular chemotherapy was significantly decreased after the intraventricular chemotherapy (ratio of 0.58/0.53/0.47) in the EV-decreased group, whereas the number of exosome markers was not significantly changed in the EV-increased group.

### 2.8. Change in miRNA Expression in Patients with LM Receiving Intraventricular Chemotherapy

As changes in EV concentration after the treatment were related to patients’ OS, we next measured the expression of a well-known onco-miRNA, miR-21, in NSCLC (adenocarcinoma). Among those above 33 patients with NSCLC, fourteen samples were available for this further analysis. The miR-21 expression measured by droplet digital polymerase chain reaction (ddPCR) (Appendix A) was normalized by EV concentration measured by NTA (Figure 5). The miR-21 expression declined (>0.2 fold change) in four out of five patients with an increased post-treatment EV concentration (mean fold change, 0.33; SD, 0.45) and was elevated in five of six patients in the EV-decreased group (mean fold change, 17.90; SD, 30.7). Among the EV concentration ‘no change’ patients, one patient showed declined miR-21 expression, and another two patients revealed no change of miR-21 expression (<0.2 fold change) (mean fold change 0.64; SD 0.55). Together, no patients showed elevated miR-21 expression among eight patients with ‘no change’ or EV-increased groups in contrast to five out of six patients in the EV-decreased group who revealed elevated miR-21 expression. Thus, the probability of elevated miR-21 expression after the treatment was significantly higher in the EV-decreased CSF samples compared with the EV-increased or ‘no change’ ones (Fisher’s exact test, *p* < 0.005).

## 3. Discussion

### 3.1. Nanoparticles in CSF

Since CSF is filtered from blood through the “blood-CSF barrier”, it contains clinically important biomolecules and extracellular vesicles of CNS disease [25,26]. Our measurement of EVs by NTA showed different mean EV size according to the different patient groups. The biggest mean EV size in LM samples might be due to a higher proportion of microvesicles and apoptotic bodies as compared with other groups, as their relative proportion of large EV (>150 nm) was higher in LM than other groups. Furthermore, the LM groups showed a significantly higher number of non-vesicular particles, which were measured by NTA as EVs (Appendix A). Although we did not identify those particles abolished after proteinase K treatment, these non-vesicular particles were mainly at a size of more than 200 nm, and they can be a result of leptomeningeal disease activity, at which cancer cells destroy the surface membrane of brain and spinal cord.

### 3.2. Differences in EV Concentration in CSF among Patient Groups by NTA

The merit of our measurement of EV concentration by NTA is that we minimized the sample manipulation and transfer before measurement (i.e., lipid membrane binding in high-resolution flow cytometry). Instead, we directly counted the number of EVs in clinically available CSF samples, requiring only two-step centrifugation to discard waste cells and debris. Our study proved that the EV concentrations of CSF is suitable for the NTA measurement range 10^6^–10^9^ particles/mL) without specific EV enrichment process, and also, it provided a reference EV concentration for unaltered CSF from hundreds of patients with various CNS diseases. Counts of EVs in CSF by NTA can be varied by camera level and detection threshold [27]. However, we used the same NTA conditions for all our samples, although some patients with inflammatory disease exhibited more EVs than the recommended particle number per frame. Thus, the relative concentration of EVs could reflect a real difference in CSF environments between patients with different CNS diseases and can also change after treatment within individual patients.

Our observation of significantly higher EV concentrations in LM patients compared with other patients could be explained by the fact that the CSF of LM patients harbors many cells, including cancer cells and white blood cells, which is known to be a major source of secreted EVs [23]. Although many researchers have found more EV secretion from cancer cells in the form of oncogenic proteins (i.e., oncosomes) [28,29], they did not measure differences in EV size between cancer and normal cells. We also could not determine whether the source of EVs was from floating cancer cells or white blood cells in this study. Among disease groups, the intracranial hemorrhage patients in the other CNS disease group exhibited the highest EV concentration of mean 1.53 × 10^9^ particles/mL followed by CNS infection. Spaull at al., who measured an EV concentration of unaltered CSF from preterm intracranial hemorrhage patients by NTA, reported 2.89 × 10^10^ to 1.31 × 10^11^ particles/mL in three samples [30]. Aside from their samples considering brisk hemorrhage in their CSF whereas our samples are not, these concentrations are comparable to each other.

### 3.3. Change in EV Concentration and miR-21 Expression in LM Patients after Intra-CSF Chemotherapy

Our observation of changes in EV concentration after chemotherapy demonstrates the possibility of conveniently analyzing patient CSF samples by commercially available NTA to provide a useful biomarker of LM treatment response. In general, increased EV secretion could reflect the removal of toxic materials such as chemotherapeutic drugs from cells, increased intercellular communication, or a modulation of the microenvironment for migration, immune regulation, or cancer metastasis [6,29,31]. Patients in our good prognostic group exhibited increased EV concentrations after treatment with methotrexate. A possible explanation could be the different mechanisms of action between drugs, such as cisplatin and adriamycin (doxorubicin), which must be eliminated by dealkylating or DNA repair enzymes, whereas methotrexate is an anti-metabolite. Another possible explanation is that our observed EVs may include not only exosomes but also microvesicles and apoptotic bodies. Thus, increased EV concentrations after treatment might reflect cellular stress and early apoptosis [26].

Unfortunately, we could not separate exosomes, microvesicles, and apoptotic bodies in this study. However, the decreased EV concentration in this study was proved to reflect a decreased concentration of exosome surface marker bearing EVs by both MACSPlex APC-conjugate beads and ExoView Tetraspanin Chips. We evaluated the implications of changes in EV concentration in a discrete cohort of LM patients with the same primary cancer of NSCLC, adenocarcinoma, who underwent the same protocol of phase II ventriculolumbar perfusion methotrexate chemotherapy. The prognostic value of miR-21 in NSCLC patients has been examined in many studies [32,33,34,35]. Our finding of markedly elevated miR-21 expression in patients with poor overall survival (and decreased EV concentration) and decreased miR-21 expression in patients with better overall survival (and increased EV concentration) after intravascular CSF chemotherapy is consistent with previous studies. Furthermore, our miR-21 data were obtained from a prospective clinical trial in which LM patients had the same primary cancer of NSCLC and received the same designated treatment protocol, proven CSF cytology, and times of CSF sampling (days 0 and 4). Thus, the prognostic meaning of miR-21 expression levels in paired samples is more valuable than that which could be obtained through spot sampling or primary cancer-only studies. 

It remains a technical problem of evaluating CSF microRNA expression level due to both the absence of endogenous control for normalization and the origin of microRNA (exosomal vs. non-exosomal) [11,12]. Recently, Parieto-Fernandez et al. performed a comprehensive measurement of CSF microRNA differentiating vesicular from non-vesicular compartment [36]. According to their study, miR-21 existed mainly in non-vesicular (CD63 negative) fraction and was nearly abolished with proteinase and RNase combined treatment. In our study, we did not use an EV enrichment procedure such as ultracentrifugation or chromatography but rather extract RNA from cleared CSF. Thus, our CSF samples naturally include non-vesicular microRNA. However, we could not explain why the miR-21 expression pattern became significantly different between the EV-decreased and EV-increased groups not by RNA amount but by NTA-measure EV concentration at this study design. A very low concentration of CSF extracellular RNA may give inconsistent results according to different extraction and RT-PCR methods [36,37]. We expect a more sophisticated EV extraction method keeping non-vesicular proteins and microRNAs from undesirable loss or damage, and solid standard RNA extraction and RT-PCR process could help to solve this question in future.

In this study, we also tracked changes in EV concentration and miR-21 expression in the CSF of LM patients after the intraventricular chemotherapy. Decreased post-treatment EV concentration and increased miR-21 expression was associated with poor prognosis in NSCLC patients with LM, who underwent intra-CSF methotrexate chemotherapy. We believe that our observation could be used as reference values for future CSF nanoparticle study and developed as a method in clinical practice to monitor the treatment response in these patients with LM.

## 4. Materials and Methods

### 4.1. Patients and Control Groups

This observational study was approved by the Institutional Review Board of the National Cancer Center (Identifier: NCC-150002, NCC-2014-0135), followed all rules and regulations regarding the protection of human subjects in clinical research, and was in accordance with the ethical guidelines outlined in the Declaration of Helsinki. All patients were signed informed consent. For patients below 18 years, the informed consent was also obtained from a parent and/or legal guardian. We analyzed CSF samples from six groups of patients. Control groups without CNS tumors were composed of patients with systemic cancer (cancer control, CC) and patients without any cancer (healthy control, HC). The second category of patients was LM from systemic cancer. The third group of patients was having parenchymal brain metastasis but no LM (brain metastasis, BM). The 4th group of patients was with primary brain tumors other than brain metastasis (brain tumor, BT). The last group of patients was from various CNS diseases but no cancer (other disease, OD). To include CC, recent imaging (e.g., computed tomography, positron emission tomography) or bone marrow reports were evaluated to detect cancerous lesions. All LM patients were cytologically diagnosed and had positive neuroimaging studies (i.e., gadolinium-enhanced brain or whole-spine magnetic resonance imaging) [17].

### 4.2. CSF Preparation and Selection for Nanoparticle Measurement

CSF samples were obtained after Institutional Review Board approval of the National Cancer Center (Identifier: NCC-2014-0135) for the purpose of identifying CSF biomarkers. CSF samples were obtained via lumbar puncture during spinal anesthesia (CC), CSF cytology examination (LM, BM, and BT), or CSF chemistry (OD). Other CSF samples were obtained from the cisternal/subarachnoid space during craniotomy (HC, BM, and BT). A total of 669 CSF samples was consecutively obtained from six groups of patients between 2014 and 2019 (Appendix A). As preliminary analysis showed that the proportions of nano-sized particles changed after treatment in LM patients (Appendix A), we excluded 197 post-treatment samples for the comparative analysis of nanoparticle distribution and concentration between patients groups. Later, 41 paired post-treatment LM samples were brought up for the evaluation of EV concentration and miR-21 expression changes after the intraventricular chemotherapy.

CSF was centrifuged within 1 h at 1500× *g* for 20 min for cells down at room temperature as storage in refrigerator (4 °C) increased nanoparticle concentration (Appendix A) in the preliminary experiment. The remainder of the samples was centrifuged again at 10,000 *g* for 30 min (cell debris removal) and kept frozen at −80 °C for genomic profiling and nanoparticle evaluation. We also confirmed that one time freezing–thawing did not alter EV concentration significantly, but EV concentration increased significantly after three times of freezing–thawing cycles (Appendix A). Hence, EV concentration measurement was done after one time freezing-thawing in all CSF samples.

### 4.3. Measurement of Nanoparticles by Dynamic Light Scattering (DLS)

CSF (40 µL) was placed in a disposable cuvette for Zetasizer Nano measurement following the manufacturer’s protocol (Malvern, Worcestershire, UK). The Zetasizer Nano system determines particle sizes (0.3 nm to 10 µm) by measuring their Brownian motion in the sample using dynamic light scattering. Of the measurement display modes, intensity particle size distribution (PSD) was chosen to represent the relative proportion of nano-sized particles.

### 4.4. Quantitative Measurement of Nanoparticles by Nanoparticle Tracking Analysis (NTA)

Pure CSF samples without dilution or concentration were used for the quantitative measurement of EV-sized particles using the NanoSight instrument (Model NTA NS300 with 642 nm red laser module, Malvern, Worcestershire, UK), which is a laser-based light scattering system that provides general nanoparticle characterization in terms of size and concentration (10^6^ to 10^9^ particles/mL). Pure CSF samples were loaded into the laser module sample chamber manually with a confirmation of no air bubbles, and an automated camera module tracked the Brownian motion of particles in the liquid sample. The in-built software (NTA 3.2, Dev. Build 3.2.16, Malvern, Worcestershire, UK) calculated the EV size and concentration of triplicate of each sample with 30 s video capture. For the consistency of the observed values, we used the sCMOS (scientific Complementary metal–oxide–semiconductor) camera type and fixed camera level at 8 of 10, and set the temperature at 25 °C. When the concentration was more than 10^9^ particles/mL, the sample was diluted, and the initial concentration was calculated according to the dilution factor.

### 4.5. CSF Concentration and Western Blotting

First, 15 mL of pre-centrifuged CSF was concentrated with Amicon^®^ Ultra 15 mL Filter (Merck, Darmstadt, Germany), and 0.2 mL of concentrate was recovered. The concentrate was measured for protein concentration using a BCA protein assay kit (Pierce Biotechnology, Rockford, IL, USA). The concentrate was diluted with sample dilution buffer (30 mM Tris-HCl, pH 6.8, 10% glycerol, 1% SDS). U-87 MG (human glioblastoma cell line) were purchased from ATCC (Manassas, VA, USA), and they were culture in RPMI-1640 supplemented with 10% fetal bovine serum and 1% penicillin/streptomycin. After washing with PBS, cells were lysed with lysis buffer (50 mM Tris-HCl, pH 7.4, 150 mM NaCl, 0.25% deoxycholic acid, 1% NP-40, 1 mM EDTA) and centrifuged at 12,000× *g* for 10 min. Cell lysate and supernatant were diluted with sample dilution buffer. The samples were used for protein expression reference. 

Samples were resolved by 10% or 15% SDS-PAGE, transferred onto nitrocellulose membrane, blocked with 5% nonfat dry milk in TBST (10 mM Tris-HCl pH 7.5, 100 mM NaCl, and 0.05% Tween 20) followed by incubation with primary antibodies against exosome membrane CD63 or CD81 (Santa Cruz Biotechnology, Dallas, TX, USA) or cytosolic protein cytochrome c or GM130 (BD Pharmingen, San Diego, CA, USA). Blots were washed and incubated with HRP-conjugated secondary antibody. Antibody complexes were visualized using an enhanced chemoluminescence Western blotting detection system (Pierce Biotechnology, Rockford, IL, USA). Relative luminescence intensities were calculated using Image J (National Institutes of Health, Bethesda, MD, USA).

### 4.6. Intact Exosome Isolation for Transmission Electron Microscopy

Exosomes in cell-free CSF were isolated using the ExoLutE exosome isolation kit (Rosetta Exosome Inc., Gyeonggi-do, Korea) according to manufacturer instructions. Briefly, thawed CSF samples were pre-filtered by a 0.45 µm syringe filter (Millex^®^, Merck, Darmstadt, Germany) to remove debris and protein aggregates. Cell-free CSF were further concentrated with 100 kDa MWCO centrifugal ultrafiltration device (Amicon^®^, Merck, Darmstadt, Germany). Intact forms of exosomes in 8 mL concentrated CSF were purified by a method that comprises exosome precipitation with metal-affinity and further purification with a spin-based size-exclusion chromatography. The purified CSF exosomes were subsequently subjected on transmission electron microscopy to determine their shapes and ultrastructures. Then, 5 µL of exosome preparation at 1 × 10^9^ particles /µL were adsorbed onto glow-discharged carbon-coated copper grids (Electron Microscopy Sciences, Hatfield, PA, USA) for 5 min. After excess liquid removal, the grid was washed 10 times with PBS and subsequently stained with 2% uranyl acetate (Ted Pella, Redding, CA, USA). The grid was finally examined in JEM 1011 microscope (JEOL, Tokyo, Japan), and images were recorded with an ES1000W Erlangshen CCD Camera (Gatan Inc. Pleasanton, CA, USA).

### 4.7. Quantitative Measurement of Extracellular Vesicles (EVs)

#### 4.7.1. MACSPlex

CSF samples after two-step centrifugation were subjected to bead-based multiplex EV analysis by flow cytometry with an MACSPlex Exosome Kit (cat. 130108813, human, Miltenyi Biotec, Gladbach, Germany) as a provided protocol [38]. Briefly, 0.5 mL of EV-containing CSF samples were processed as follows: Samples were added to 0.5 mL of MACSPlex buffer (MPB), 15 μL of capture beads, and 15 uL of detection reagent mixture (APC-conjugated anti-CD9, anti-CD63, and anti-CD81 antibodies) and incubated overnight using an orbital shaker (450 rpm) protected from light. To wash the samples, 500 μL of MPB was added to each tube and centrifuged at 3000× *g* for 5 min, and the supernatant were removed. After washing 3 times, samples were resuspended to 0.5 mL of MPB buffer and transferred to a 5 mL round-bottom tube (cat. 352235, BD Bioscience, San Jose, CA, USA) and analyzed with a BD LSRfortessa™ (BD Biosciences, San Jose, CA, USA). For setting up the instruments, the Exosome setup beads were used, and the flow cytometry data were analyzed using FlowJo software (v10, BD, San Jose, CA, USA). The median fluorescence intensity (MFI) values of APC for CD9, CD63 and CD81 bead populations were used to determine the expression of exosome markers.

#### 4.7.2. Exoview

Quantitative measurement of EVs was tested with ExoView Tetraspanin kits (NanoView Bioscience, Boston, MA, USA) as previously decribed [39]. CSF was diluted in PBS with 0.05% Tween-20 (PBST) and incubated on the ExoView Tetraspanin Chip (EV-TC-TTS-01) placed in a sealed 24-well plate for overnight at room temperature. After washing with PBST three times, chips were incubated with ExoView Tetraspanin Labelling ABs (EV-TC-AB-01) that consist of anti-CD81 Alexa-555, anti-CD63 Alexa-488, and anti-CD9 Alexa-647. The antibodies were diluted 1:5000 in PBST with 2% BSA. The chips were incubated with 250 μL of the labeling solution for 2 h. Then, the chips were washed once in PBST, three times in PBS followed by rinsing in filtered DI water and dried. Then, the chips were imaged with the ExoView R100 reader (Nanoview Bioscience, Boston, MA, USA) using the ExoScan 2.5.5 acquisition software. Then, the data were analyzed using ExoViewer 2.5.0 with sizing thresholds set to 50 to 200 nm diameter.

### 4.8. Extracellular RNA Extraction from CSF

We used mirVana PARIS (Ambion, Mulgrave, Victoria, Australia) to extract extracellular RNA from cleared CSF according to the manufacturer’s instructions. Briefly, 0.5 mL CSF was gently mixed with 2× denaturing solution, acid-phenol:chloroform was added, and the mixture was centrifuged at 10,000× *g* for 5 min to separate the aqueous phase. After phase separation, a 1.25 volume of 100% ethanol was added to the aqueous phase, and the solution was loaded into the column and centrifuged at 12,000× *g* for 30 s. Extracted RNA was washed and eluted with 25 µL (95 °C) RNase-free water. The purity (A260/280 and A260/230 ratio) and concentration of dissolved RNA samples were measured using a Nanodrop spectrophotometer (ThermoFisher Scientific, Waltham, MA, USA).

### 4.9. miRNA Digital Droplet Polymerase Chain Reaction (ddPCR)

To evaluate differences in miRNA expression between LM and normal CSF samples, we measured the expression of hsa-miR-21-5p by ddPCR. First, 2 µL purified total RNA was reverse-transcribed into cDNA using a Taqman advanced miRNA cDNA Synthesis Kit (Applied Biosystems, Foster City, CA, USA) according to the manufacturer’s instructions. A total of 5.5 µL undiluted cDNA was mixed with 2× ddPCR Supermix (Bio-Rad Laboratories, Inc., Hercules, CA, USA) and Taqman advanced miRNA Assay probes (Applied Biosystems, Foster City, CA, USA). Each PCR sample was partitioned into up to 20,000 nl-sized droplets by a Bio-Rad QX299 droplet generator. Then, 40 µL of each PCR mixture underwent the following cycling protocol: 95 °C for 10 min (DNA polymerase activation), followed by 40 cycles of 94 °C for 30 s (denaturation) and 60 °C for 1 min (annealing), followed by post-cycling steps of 98 °C for 10 min (enzyme inactivation) and an infinite 4 °C hold. The amplified PCR product of the nucleic acid target in the droplets was quantified in the FAM channel using a QC200 Droplet Reader (Bio-Rad Laboratories, Inc., Hercules, CA, USA) and analyzed using QuantaSoft v.1.7.4.0917 software (Bio-Rad Laboratories, Inc., Hercules, CA, USA).

### 4.10. Statistical Analysis

Categorical data were analyzed using Chi-square, Fisher’s exact, or Mann-Whitney U tests as appropriate. Continuous data were analyzed using Student’s t-tests or analysis of variance (ANOVA) with parametric scheffe’s test and nonparametric Dunn’s tests for post hoc comparisons. The Kolmogorov-Smirnov test was used for the comparison of Ecdf analysis. A two-tailed *p*-value < 0.05 was considered statistically significant. Statistical analyses were performed using SPSS 18.0 (SPSS, Chicago, IL, USA), GraphPadPrism 6 (GraphPadPrism Software, La Jolla, CA, USA), and R software 3.6.0 (Boston, MA, USA).

## 5. Conclusions

We measured nanoparticles in human CSF from 472 controls and patients including leptomeningeal metastasis (LM), which has neither effective treatment nor biomarkers for disease progression or treatment response, by both Dynamic Light Scattering and Nanopaticle Tracking Analysis. We analyzed the difference of size and concentration according to different CNS disease status. In addition, we found the change of EVs concentration after intra-CSF chemotherapy in patients with LM to correlate with patients overall survival, and the expression level of onco-microRNA (miR-21) was in inverse proportion to EV concentration change in non-small cell lung cancer. Our measurement of combined EVs concentration and onco-miR for LM chemotherapy could be performed in moderately equipped institution and help physicians monitor this possible neurotoxic treatment.

## Figures and Tables

**Figure 1 cancers-12-02745-f001:**
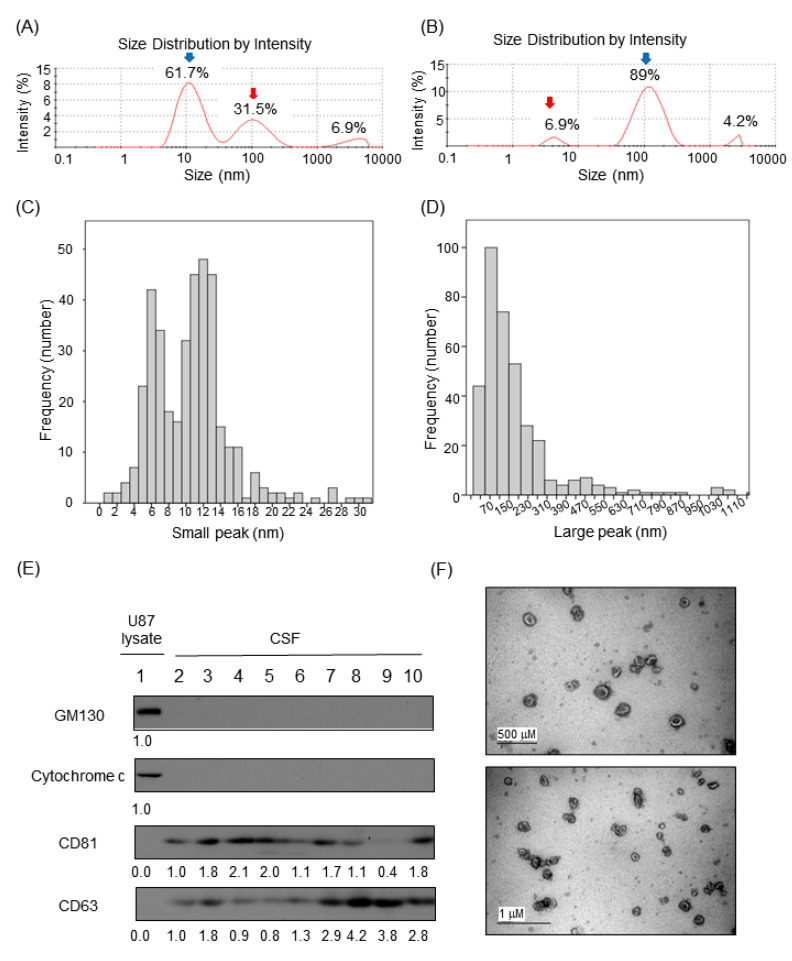
Identification of extracellular vesicle markers from CSF. (**A** and **B**) Representative diagram of observed peaks of nanoparticles in human cerebrospinal fluid by Dynamic Light Scattering (DLS) using the Zetasizer Nano system. (**A**) Healthy control and (**B**) leptomeningeal metastasis (LM). (**C** and **D**) Histogram depicting the two peaks in human CSF (n = 472) observed by Dynamic Light Scattering (DLS) using the Zetasizer Nano system. (**C**) The small peak was measured at a mean of 10.5 nm and (**D**) the large peak was observed at a mean of 176 nm. (**E**) Western blot of human CSF for exosomal markers. (**E**) Western blot of human CSF for exosomal markers. Golgi matrix protein of 130 kDa (GM130) and cytochrome c were used for the cytosolic organelle markers. U87 glioma cell extract was used as positive control. CD81 and CD63 were used for the exosomal membrane protein markers. Relative band intensities were written under each immunoblot. Samples in lane 2–6 were in LM group. Samples in lane 7–8 were in the brain metastasis (BM) group. Samples in lane 9–10 were in the other disease (OD) group. Relative band intensities were written under each immunoblot. Uncropped Western blots are shown in Appendix A. (**F**) Transmission Electron Microscopy showing bi-membranous vesicles of purified CSF exosomes. scale bar; 500 nm (up), 1 μm (down).

**Figure 2 cancers-12-02745-f002:**
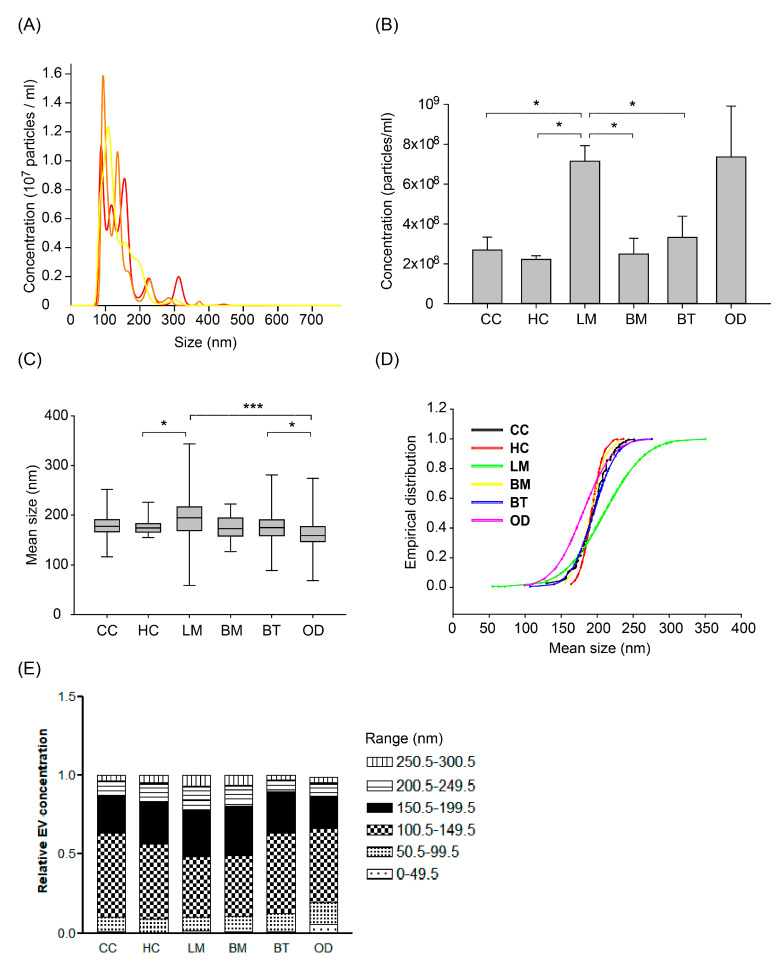
The number of extracellular vesicles (EVs) measured by Nanoparticle Tracking Analysis using the Nanosight NS300 system according to patients groups. (**A**) Representative image of NTA measurement (repeated 3 times per one sample and an average value was provided). (**B**) Extracellular vesicle (EV) concentration according to patient groups. Different EV concentrations among groups of ‘Other disease’ according to different disease of multiple sclerosis, intracranial hemorrhage, and central nervous system (CNS) infection (see details in Appendix A) (ANOVA, Dunn’s multiple comparison, * *p* < 0.05). (**C**) Box plot of size distribution of EVs according to patient groups. The thick bar represents the mean value, and the box denotes the quartile range. (ANOVA, Dunn’s multiple comparison, * *p* < 0.05, *** *p* < 0.001) (**D**) Empirical cumulative distribution plot of mean size of extracellular vesicles. (The empirical cumulative distribution function (ECDF) was generated by the Ecdf function in R, Kolmogorov-Smirnov tests were performed to compare the significance between groups) (**E**) The proportion of EV size by tentative size interval of 50 nm (0–300 nm). Abbreviations: CC, cancer control; HC, healthy control; LM, leptomeningeal metastasis; BM, brain metastasis; BT, brain tumor.

**Figure 3 cancers-12-02745-f003:**
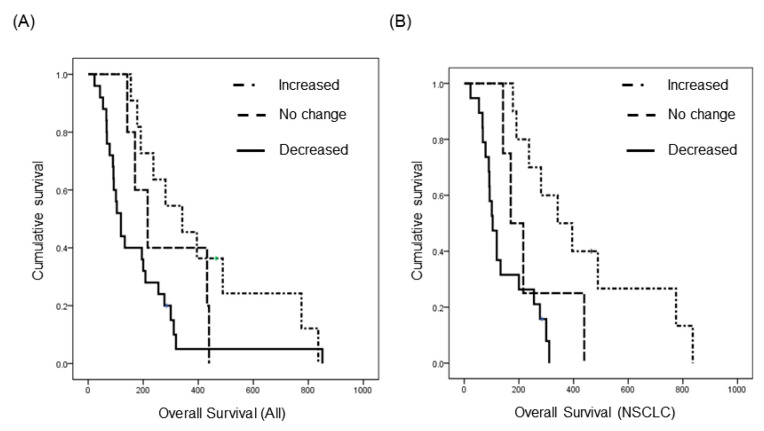
Kaplan–Meier overall survival analysis of patients who received the same intra-CSF chemotherapy for LM according to the extracellular vesicle concentration change after the treatment. (**A**) All 41 patients with various primary cancer (*p* < 0.05), and (**B**) non-small cell lung cancer patients (n = 33, *p* < 0.001).

**Figure 4 cancers-12-02745-f004:**
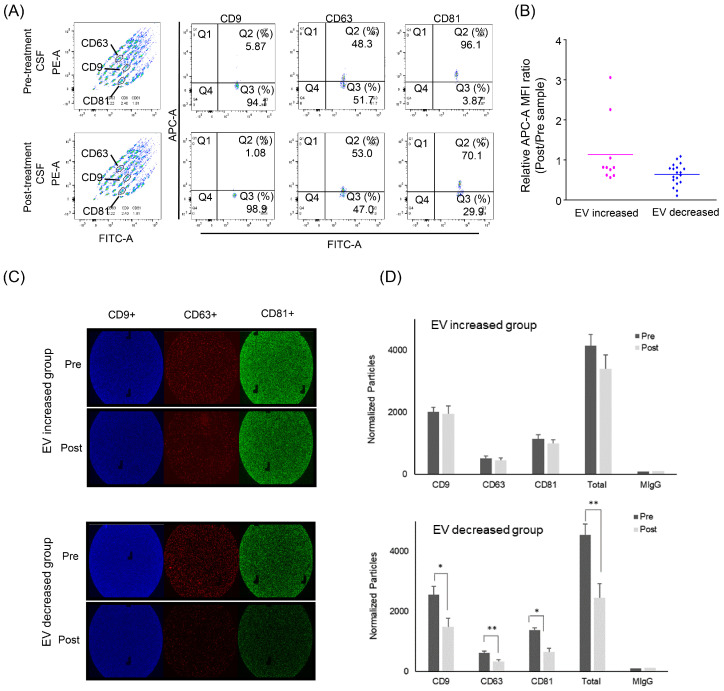
Evaluation of exosome concentration change after intraventricular chemotherapy by surface markers. (**A**) Flow cytometry using the MACSPlex Exosome Kit. (**B**) The relative expression of EV markers was determined by the mean fluorescence intensity (MFI) of APC-conjugated capture antibody. In the EV-decreased group, the exosome markers were significantly reduced after intraventricular chemotherapy (ratio 0.64, *p* < 0.001), whereas increased groups showed no significant exosome concentration change (ratio 1.13). (**C**) ExoView Tetraspanin Chip revealing exosome surface markers (CD9 in red, CD63 in green and CD81 in blue). (**D**) The change of each exosome marker after the intraventricular chemotherapy showed a significant decrease of all exosome markers in the EV-decreased groups (*p* * < 0.05, *p* ** < 0.005). Tx: Treatment.

**Figure 5 cancers-12-02745-f005:**
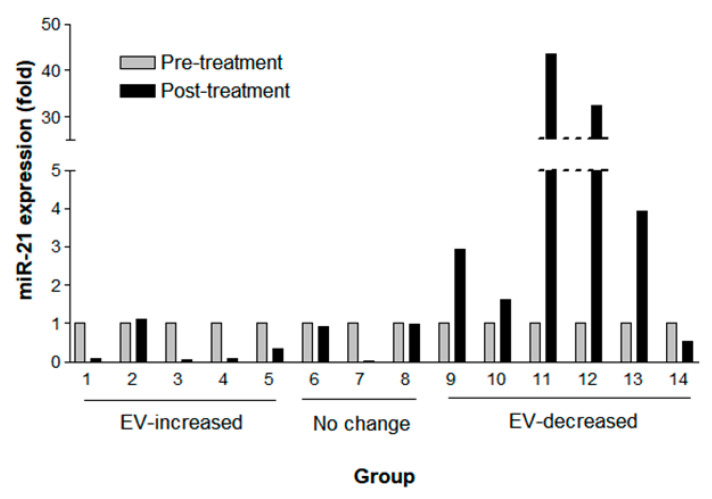
Fold change of miR-21 expression by digital droplet PCR paired pre-treatment (gray column) and post-treatment (black column) CSF samples according to EV changes of increased (n = 5), ‘no change’ (n = 3), and decreased (n = 6) EV concentration after the intraventricular chemotherapy. CSF extracellular miR-21 was detected by digital droplet PCR and normalized with EV concentration.

**Table 1 cancers-12-02745-t001:** Clinical characteristics of cerebrospinal fluid (CSF) sample analysis (n = 472).

Groups	Total	Cancer Control (n = 100)	Healthy Control (n = 73)	LM (n = 150)	Brain Metastasis (n = 28)	Brain Tumors (n = 72)	Other Disease of CNS (n = 49)
Gender							
Male (%)	215 (45%)	70 (70%)	31 (43%)	53 (36%)	13 (46%)	36 (50%)	12 (25%)
Female (%)	257 (55%)	30 (30%)	42 (57%)	97 (64%)	15 (54%)	36 (50%)	37 (75%)
Median age (range)	48 (0.2–90)	17 (1.5–90)	59 (20–80)	52 (2.0–79)	55 (7–79)	17 (1.9–80)	40 (0.2–83)
Combined disease (%)		Leukemia (62) Bladder ca. (8) Bone tumor (7) Lymphoma (4) Breast ca. (3) Colon ca. (3) Prostate ca. (2) Cholangioca. (2) Melanoma (2) Others (7)	Unruptured An. (43) Moyamoya ds. (12) Hydrocephalus (5) ICA stenosis (4) BPH (2) Head trauma (2) Fibromatosis (2) Others (3)	NSCLC (64) Breast ca. (35) Glioma (15) Non-glial BT (10) SCLC (4) Stomach ca. (4) PCNSL (2) MUO (2) melanoma (2) Others (16)	NSCLC (16) Breast ca. (6) HCC (1) Melanoma (1) Ovarian ca. (2) Sarcoma (1) SCLC (1)	Glioma (19) Medulloblastoma (15) Germ cell tumor (9) Non-glial MBT (6) Ependymoma (7) Pituitary adenoma (6) Other benign BT (6) Neurinoma (4)	MS (25) ICH (13) Infection (9) Other AID (2)

Abbreviations: AID, autoimmune disease; An, aneurysm; BPH, benign prostate hypertrophy; BT, brain tumor; HCC, hepatocellular carcinoma; ICA, internal carotid artery; ICH, intracranial hemorrhage; LM, Leptomeningeal metastasis; MBT, malignant brain tumor; MS, multiple sclerosis; MUO, malignancy of unknown origin; NSCLC, non-small cell lung cancer; PCNSL, primary central nervous system lymphoma; SCLC, small cell lung cancer. Numbers in parenthesis are vertical proportion.

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
