# Peer review of "Nanoparticles in 472 Human Cerebrospinal Fluid: Changes in Extracellular Vesicle Concentration and miR-21 Expression as a Biomarker for Leptomeningeal Metastasis"

_cancers, 2020, doi:10.3390/cancers12102745_

Round 1
Reviewer 1 Report
In this paper “Nanoparticles in 472 Human Cerebrospinal Fluid: Changes in Extracellular Vesicle Concentration and miR-21 Expression as a Biomarker for Leptomeningeal metastasis.” Kyue-Yim Lee and coworkers proposed the new method that might be useful to diagnose or predict outcomes of leptomeningeal metastasis. The suggested method is based on monitoring of changes in the concentration of extracellular vesicles in cerebrospinal fluid. The conducted experiments are very interesting however, there are some issues that need to be addressed before a manuscript can be recommended for publication.
1). First at all, the authors showed that 61% of LM patients after intraventricular chemotherapy showed a decrease in EV concentration which was correlated with a decrease in overall survival. Did I understand correctly that VLP chemotherapy with methotrexate reduces patients' survival rather than prolongs them?
2). Next, in the Materials and Methods section, there are some deficiencies or gaps. Whether the authors used any criteria for excluding patients from the study? Did the authors check the stability of the CSF at -80oC? The term GM130 requires some explanation. Which exactly means specifying a “clarified CSF”? What was the average quality of the RNA extracted (A230/280 ratio)?
3). Some graphs, especially Figure1A-D, Figure 4A, are difficult to read. They should be enlarged enough to read the markings and the font. Moreover, the graph in the Figure 2D is completely unreadable. The form of presenting the results should be rethought.
4). In Figure 1E. The description of the figure is misleading. The authors mention upper 3 rows (in line 241), but I only see two (GM130 and cytochrome c). Also, where exactly is the control of HT-29 supernatant. In addition, why did the authors decide to use the colon cancer cell line as a positive control for exosomes? In addition, what extracts hide under numbers 2-10 of CSF?
5). In 3.5 section, in the identification of protein aggregates in CSF nanoparticles, the OD fraction should also be examined, especially since as shown by the results in Figure 2B, they have a similar number of EVs as the LM fraction.
6). On line 226, Figure 1C-B is written. Is this correct?
Author Response
"Please see the attachment."

Reviewer 2 Report
First, thank you for giving me the opportunity to review this manuscript. English is not my first language, so I am not able to correct all grammar mistakes (if present)
This study analyzes nanoparticules in large number of samples of human CSF in different disease conditions. The question raised is clear, and the way the authors respond to this question is accurate. The methods are very well described and are also accurate. The experiments are well conducted and I do not have major comments. Although the results are not always clear-cut between groups, this manuscript is worth publication.
I have some minor comments:
Supp figure 5 : I think that error bars are not well located on supplementary figure 5C
Statistical methods needs to be expanded : the authors use statistical tests in the result section that are not described in the methods section.
I do not have major comments
I did not detect plagiarism. I do not have Ethic concern. I did not detect image manipulation
My suggestion is: Accept after minor revision
Author Response
"Please see the attachment."

Round 2
Reviewer 1 Report
I appreciate the authors put much effort to fix the issues that were mentioned in the original review. The corrections implemented by the Authors improve the quality of the manuscript.
Author Response
We appreciate reviewer's comments and suggestions